# MassWateR: Improving quality control, analysis, and sharing of water quality data

**Marcus W. Beck** [iD][1]*, **Benjamen Wetherill**[2], **Jillian Carr**[3]

**1** Tampa Bay Estuary Program, St. Petersburg, Florida, United States of America, **2** ACASAK Consulting, Boston, Massachusetts United States of America, **3** Massachusetts Bays National Estuary Partnership, Boston, Massachusetts, United States of America

* mbeck@tbep.org

## Abstract

The long-term protection and restoration of aquatic resources depends on robust monitoring data; data that require systematic quality control and analysis tools. The *MassWateR* R package facilitates quality control, analysis, and data sharing for discrete surface water quality data collected by monitoring programs of various size and technical capacity. The tools were developed to address regional needs for programs in Massachusetts, USA, but the principles and outputs can be applicable to monitoring data collected anywhere. Users can create quality control reports, perform outlier analyses, and assess trends by season, date, and site for more than 40 parameters. Users can also prepare data for submission to the United States Environmental Protection Agency Water Quality Exchange, thus sharing data to the largest water quality database in the United States. The automated and reproducible workflow offered by *MassWateR* is expected to increase the quantity and quality of publicly available data to support the management of aquatic resources.

**Data Availability Statement:** The R package is hosted on CRAN at https://cran.r-project.org/package=MassWateR. The source code for the package is available on GitHub at https://github.com/massbays-tech/MassWateR.

## 1. Introduction

Water quality measurements provide the foundation of environmental monitoring programs designed to protect or restore aquatic resources. In the United States, these programs are broadly guided by the federal Clean Water Act (33 USC § 1251 et seq.) with the singular goal of restoring and maintaining the chemical, physical, and biological integrity of the nation's surface waters. Similarly, the Water Framework Directive provides a legislative foundation for the protection of aquatic resources in member states of the European Union (Directive 2000/60/EC of the European Parliament and of the Council of 23 October 2000) [1]. Numeric standards that define critical thresholds for protecting recreational, aquatic life, industrial, navigational, and consumptive uses of the resource are often established by government agencies, such that exceedances identified from water quality measurements require additional regulatory action to ensure compliance. These standards and other regulatory assessments as applied at the state-level use information from long-term monitoring datasets [2,3], or data collected *ad hoc* from multiple assessment endpoints [4–6], where the former is atypical for most surface water bodies. Many state or regional institutions that assess water quality rely on decentralized

**Funding:** This work was supported by an Exchange Network grant from the US Environmental Protection Agency awarded to the Massachusetts Bays National Estuary Partnership, Grant No. OS-84029801-0. The funders had no role in study design, data collection and analysis, decision to publish, or preparation of the manuscript.

**Competing interests:** The authors have declared that no competing interests exist.

data sources, often combining datasets from local watershed groups or participatory science programs rather than a single database that contains adequate coverage of all areas of interest [7,8]. Use of these monitoring data in a regulatory context is not possible unless standard operating procedures are adopted and the data fulfill quality control (QC) requirements [e.g., 9–11].

Monitoring data of sufficient quantity and quality are critical to ensure precise and accurate representation of environmental conditions. A significant bottleneck in the use of monitoring data for environmental assessment of surface waters is the ability to clearly and efficiently indicate that the data fulfill applicable QC checks for regulatory applications or inclusion in a consolidated database [12]. Common QC checks for *in situ* field measurements or concentrations measured in the laboratory may include 1) comparison of the precision between replicate samples (duplicates), 2) accuracy of a sample relative to a known concentration (spikes or instrument checks), and 3) evaluation of the measurement from an empty or blank sample (blanks) [13]. An adequate number of QC samples must also be included in the dataset as a measure of "completeness". These checks are often compiled in a single report for review by appropriate regulatory agencies. For example, precision of duplicate samples for a given parameter must not vary more than 5% and at least 10% of the data should be dedicated to these checks as a measure of completeness. For local monitoring groups that lack the resources to develop robust and repeatable workflows, QC reports are often prepared manually before submitting the data for potential use by agencies. This process is time-consuming and prone to errors, often limiting the amount of useful information for regulatory assessments or submitted to formal databases. As an example, dozens of community-driven environmental organizations in Massachusetts, USA, collect water quality data, yet government agencies are only able to access and utilize a fraction of those datasets. The Massachusetts Bays National Estuary Partnership funded the development of *MassWateR* described herein to build capacity among water monitoring programs to generate and share high quality data. This capacity need is not specific to Massachusetts, thus, the tools were developed with transferability in mind.

The use of R [14] with document generation systems offered through packages like *knitr* [15] and *rmarkdown* [16] can generate QC reports that follow a standard format for review by regulatory agencies. These tools can also be used to format water quality data for submission to state or national water quality databases, such as the Water Quality Portal (WQP) database maintained by the US Environmental Protection Agency (USEPA) and contributed to via the Water Quality eXchange (WQX). This database is the largest source of monitoring data in the United States that includes information on hydrologic conditions and chemical, physical, and biological measurements from surface waters. Further, many environmental resource managers have the need to analyze status and trends in monitoring data and R packages such as *ggplot2* [17] and others in the *tidyverse* [18] offer useful tools to synthesize and visualize results. Integrating this functionality into a single package is expected to have wide ranging utility for anyone collecting surface water data and is likely to improve the quality and insights obtained from these data.

To our knowledge, there are no existing R packages on the Comprehensive R Archive Network (CRAN) that can be used to facilitate QC of water quality data, nor are any available that facilitate submission to existing databases. However, there are several that can be used to retrieve and analyze data from existing sources (see the CRAN Hydrology Task View). In particular, the *dataRetrieval* R package [19] has been used widely to retrieve data from the WQP, which is the interface for accessing data submitted using WQX. This package leverages a robust API to query existing water quality data in standardized format provided by the WQP. The *TADA* R package [20] is also currently under development as a resource for compiling and evaluating data from the WQP. *TADA* is similar to *dataRetrieval* in that it can be used for importing data, but the package is also expected to provide more comprehensive methods for

cleaning, filtering, and processing data using the rich qualifier codes provided by WQP. As such, data retrieval using existing web services is much simpler than data submission, as data formatting requirements do not apply when retrieving data. Developing a robust tool that can facilitate the upload of data to WQX, in addition to streamlining QC processes, would further the value of packages like *dataRetrieval* and *TADA* by increasing the amount of data that can be accessed through the WQP. The *MassWateR* package was developed to provide this benefit.

Other software platforms outside of the R environment provide various services for quality control of water quality data. For example, the Aquarius software environment is a proprietary resource for managing hydrologic and water quality data and is used by several private and public institutions. Functionality is provided to pre-process and synthesize multiple data streams, manually correct erroneous values, and to visualize results for decision-making. A graphical user interface is provided to access the various features of the software, as compared to a programmatic approach for building custom routines. The software is not open-source, which implicitly limits its development to a core set of maintainers and is not freely accessible to the broader community [e.g., 21]. Similar platforms are available from companies that manufacture data loggers (e.g., YSI, HOBO), all of which are specific to the monitoring equipment and not broadly transferable. Alternative publicly funded software and data services are provided by the Consortium of Universities for the Advancement of Hydrologic Science (CUAHSI). Services provided by CUAHSI include data discovery, archiving, cloud computing, and analysis, with many of the services available in open source environments (e.g., Python). However, the services provided by CUAHSI are meant to address a variety of different resources and use cases, none of which are immediately related to quality control reporting and data submission to public institutions. As such, specific software solutions to address these needs and that leverage existing tools available in a rich open-source environment are clearly needed.

This paper describes the *MassWateR* package developed to improve how environmental professionals perform quality control, analysis, and sharing of monitoring data for surface waters. The regional focus of the package is for monitoring data collected in Massachusetts, USA, with QC reports submitted to the Massachusetts Department of Environmental Protection (MADEP) and data submitted to the national WQP database. Although the initial conception of *MassWateR* was to address regional needs in Massachusetts, there is nothing specific in the package that prevents its use outside of the state as the QC checks and analyses follow routine and commonly used methods for data collected elsewhere. As such, this paper is written with emphasis on how the tools are broadly applicable to anyone interested in improving efficiency and reproducibility of QC checks, in addition to analysis of water quality data and submission to WQX as the largest source of water quality monitoring data in the US.

## 2. Requirements for use

Users can engage with *MassWateR* to achieve different goals. This design was intentional based on likely differences in needs among the user community. Although increasing data submission and facilitating QC reporting was the primary goal, we also assumed that users may not want to do both. That is, state institutions require QC reporting for regulatory assessments, whereas data submission to WQX may be a separate process. Users may also simply have a need to explore trends or to summarize their data, while also wanting to extend these analyses beyond *MassWateR* using additional R packages. Fig 1 demonstrates how a user may apply the functions in *MassWateR* once the required data are imported. The functions allow a user to engage with their data in several ways. The first step, QC screening, is often iterative as a user can modify parts of the raw data based on messages from the data import functions or checks

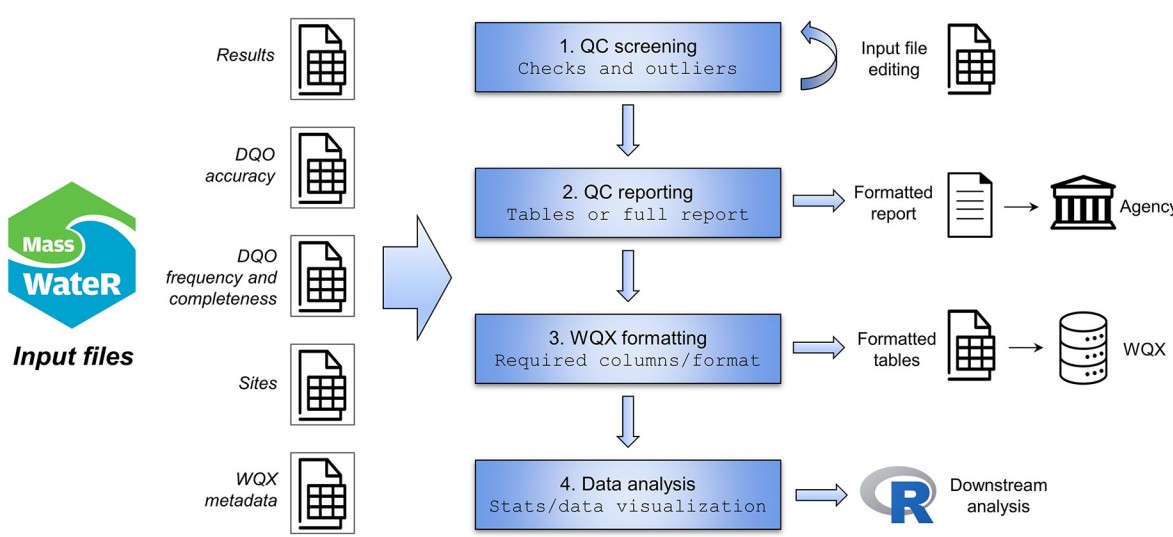

**Fig 1. Workflow demonstrating how a user could engage with the MassWateR package.** DQO: Data Quality Objectives; QC: Quality Control; WQX: Water Quality Exchange.

for outliers. The second step can be used to create a QC report for submission to a regulatory agency. The third step is data analysis and visualization, using MassWateR functions and downstream analysis with additional R packages and functions. The fourth and final step can create a formatted table for WQX submission.

No matter the user need, all data inputs to *MassWateR* must follow a strict format. Developing a workflow to accommodate data inputs from the dozens of potential users from several organizations that use different data formats would have been impractical. As such, the primary limitation to using the package is to adhere to the formatting requirements for all input files. Several resources are provided on the package web page to assist users in formatting their data. These resources included several training activities that were conducted during package development and templates demonstrating the appropriate format and rationale. The trainings, *pkgdown* website [22], and Community of Practice forum were also implemented so that learning R was not a significant limitation for using the package.

The required data files for using *MassWateR* are shown in Table 1, including how each file applies to the workflow steps in Fig 1. The files are imported into R using specific read functions with relevant checks, explained in the next section. These checks verify multiple

**Table 1. File requirements for using MassWate R.** Check marks indicate which file is required for each part of the MassWateR workflow. DQO: Data Quality Objective; QC: Quality Control.

| Formatted file | Description | 1. QC screening | 2. QC reporting | 3. Data analysis | 4. WQX formatting |
|---|---|---|---|---|---|
| Results | Water quality results organized by sample location and date | ✓ | ✓ | ✓ | ✓ |
| DQO accuracy | Summary of data quality objectives that describe quality control accuracy for data in the results file | ✓ | ✓ | ✓ | ✓ |
| DQO frequency and completeness | Summary of data quality objectives that describe quality control frequency and completeness measures for data in the results file | ✓ | ✓ | | |
| Sites | A site metadata file, including location names, latitude, longitude, and additional grouping factors for sites in the results file | ✓ | | ✓ | ✓ |
| WQX metadata | A wqx metadata file required for generating output to facilitate data upload to WQX | | | | ✓ |

requirements outlined in the template files, with informative errors or warnings returned to the console to prompt the user on the required action to remedy a formatting issue. The largest input file required for all parts of the workflow in Fig 1 is the results file. This file includes all water quality monitoring data to be used with the package and the recommendation is that the results data include a year of sampling as reporting typically follows an annual cycle. The results file design also considered WQX requirements to facilitate data submission. As such, the formatting requirements for the results file are the most burdensome for potential users and additional functions are available to assist in this effort.

The following sections describe the basic approach to using functions in *MassWateR* for any of the processes in the workflow. The naming convention for the functions is meant to provide users with an intuitive format for understanding what each function does and the step of the workflow for which the function applies. Although there are some exceptions to this nomenclature, the general format includes a prefix for each function as follows. Each prefix also includes MWR to avoid namespace conflicts with other packages (e.g., readMWR).

- read: Read input files

- check: Check input files for formatting issues, used internally to the read functions

- form: Format input files for downstream functions, used internally to the read functions once the checks have passed

- anlz: Analyze imported data

- tab: Create formatted tables for QC analysis

- qc: QC functions for summarizing QC results, used internally to the table functions

- util: Various utility functions that accomplish routine tasks, possibly useful as standalone functions

Additionally, functions may often include a suffix that describes the relevant file used as input or otherwise evaluated in a downstream function.

- results: The results input file

- acc: The data quality objectives file for accuracy

- frecom: The data quality objectives file for frequency and completeness

- sites: The site metadata file

- wqx: The WQX metadata file

## 3. Package workflow

### 3.1 Read and check files

The primary task of the read functions is to ensure all imported files follow the required format for the package. Excel files are the expected format for all inputs and the read functions use the read_excel function from the *readxl* package [23]. The read functions do very little other than import the file—once the file is imported it is immediately passed to one of the relevant check functions inside the read function. There are several checks for each type of input file, with the number of checks increasing based on the complexity of the input file. Each check is printed to the R console on completion, whereas an error is returned at the first instance of a failed check, at which point the function exits. The error will typically indicate which parts of the

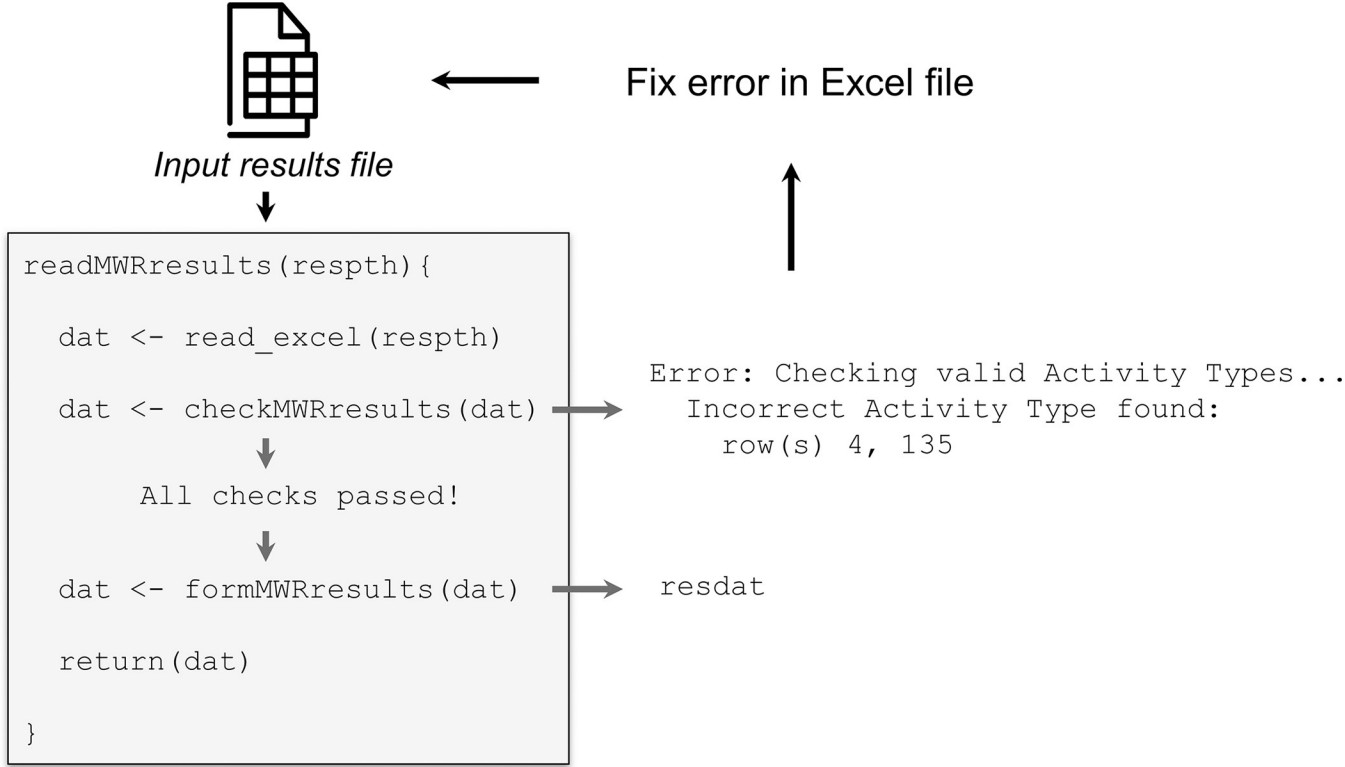

**Fig 2. Pseudocode demonstrating the iterative process of importing a required data file for MassWateR.** All read functions import an Excel file from the user-specified path (e.g., respth) and the imported file is then passed to a check function. The function exits if an error is encountered, allowing the user to manually fix the identified error and then import again. After all checks are passed, a formatting function is applied to correct minor issues (e.g., standardize date format as YYYY-MM-DD) and the final data object is returned (e.g., resdat).

input file need to be changed to rectify the issue, often indicating a specific cell in the Excel file that requires attention. As such, the workflow is intended to be iterative, where a user imports a file, receives an error, manually changes the input file in Excel, then imports the data again until all checks pass (Fig 2). Again, this design was intentional as many monitoring agencies and groups organize data differently and a standard input format for the package was the best option to accommodate all potential users. This may also encourage future standardization among monitoring groups for how data are maintained to ease formatting challenges to using *MassWateR*. A user only needs to format their data once to use the package.

A correctly formatted input file would be imported as follows, with the messages in the console indicating the checks that were performed and that all checks were successful. Below demonstrates what would be shown for the results file using an example dataset included with the package. A total of fifteen checks are applied to the results file (described in detail in the help documentation and vignettes).

```
library(MassWateR)
# import results data
respth <- system.file("extdata/ExampleResults.xlsx", package =
"MassWateR")
resdat <- readMWRresults(respth)
#> Running checks on results data...
#> Checking column names... OK
#> Checking all required columns are present... OK
```

```
#> Checking valid Activity Types... OK
#> Checking Activity Start Date formats... OK
#> Checking depth data present... OK
#> Checking for non-numeric values in Activity Depth/Height Measure...
OK
#> Checking Activity Depth/Height Unit... OK
#> Checking Activity Relative Depth Name formats... OK
#> Checking values in Activity Depth/Height Measure > 1 m / 3.3 ft... OK
#> Checking Characteristic Name formats... OK
#> Checking Result Values... OK
#> Checking QC Reference Values... OK
#> Checking for missing entries for Result Unit... OK
#> Checking if more than one unit per Characteristic Name... OK
#> Checking acceptable units for each entry in Characteristic Name...
OK
#> All checks passed!
```

The following shows a typical error that might be returned if a check fails when importing the results file. The resdat object is an imported results file that has passed all checks, but incorrect entries are added to the chk object to demonstrate the error that is returned if a user would have attempted to import this file. The checkMWRresults() function is run inside the readMWRresults() function and runs the checks (e.g., the column names are correct, all required columns are present, activity types are valid, etc.), but then stops when invalid activity types in the Activity Type column are found. In this example, a user would need to change the entries in rows 4 and 135 of the Activity Type column in their Excel file to fix the issue and import the file again as in Fig 2. The online vignette specifies the valid entries.

```
chk <- resdat
chk[4, 2] <- "Sample"
chk[135, 2] <- "Field"
checkMWRresults(chk)
#> Running checks on results data...
#> Checking column names... OK
#> Checking all required columns are present... OK
#> Error: Checking valid Activity Types...
#> Incorrect Activity Type found: Sample, Field in row(s) 4, 135
```

The readMWRresultsview() function is also available to assist with troubleshooting formatting issues for the results file. This function exports an Excel file that shows the unique values that are found in each column to allow a user to quickly see potential incorrect entries or typos. The output is similar to running the following:

```
apply(readxl::read_excel('path/to/results.xlsx'), 2, unique)
```

However, this approach returns unique column entries in the R console as a list object. Users may be more comfortable evaluating the external file created by readMWRresultsview() to troubleshoot formatting problems.

After all file format checks are completed and errors fixed, a standard approach for using *MassWateR* is to import all required files and save them as a list of named data frame objects that can be used by nearly all the package functions. This prevents the need to identify which input datasets are needed for each function, although the latter approach could be used because arguments for individual input files are also provided in all functions. In the latter

case, a path or data object can be used as input for each file. For the former approach, the beginning of a script for using the package could appear as follows. Example files included with the package are imported (see the S1 File for the example data quality objective files for accuracy, frequency and completeness), whereas a user will specify paths to their own files.

```
# import results data
respth <- system.file("extdata/ExampleResults.xlsx", package =
"MassWateR")
resdat <- readMWRresults(respth)
# import accuracy data
accpth <- system.file("extdata/ExampleDQOAccuracy.xlsx", package =
"MassWateR")
accdat <- readMWRacc(accpth)
# import frequency and completeness data
frecompth <- system.file("extdata/ExampleDQOFrequencyCompleteness.
xlsx", package = "MassWateR")
frecomdat <- readMWRfrecom(frecompth)
# import site data
sitpth <- system.file("extdata/ExampleSites.xlsx", package =
"MassWateR")
sitdat <- readMWRsites(sitpth)
# import WQX metadata
wqxpth <- system.file("extdata/ExampleWQX.xlsx", package =
"MassWateR")
wqxdat <- readMWRwqx(wqxpth)
# a list of input data frames
fsetls <- list(res = resdat, acc = accdat, frecom = frecomdat,
sit = sitdat, wqx = wqxdat)
```

The object fsetls can then be used as input to downstream functions.

A final note about the data inputs is that many types of water quality measurements can be included for analysis, although the package is currently limited to working with discrete samples as compared to data from continuous monitoring equipment. The paramsMWR dataset included with *MassWateR* provides a lookup table of the parameters that can be used with the package (see S1 File). On data import, this dataset is referenced to ensure that only relevant parameters are included and that appropriate units of measurement are provided. The dataset includes 43 different parameters, each with multiple valid units of measurement. Additionally, only one unit of measurement is allowed per parameter, which was an intentional design so that tedious functions for converting between dozens of units of measurement did not need to be created during package development. It is not an unreasonable expectation for users to provide only one unit of measurement per parameter. Laboratories typically have a standard reporting format based on the same methodology or instrument used to measure water quality parameters. Both the number of parameters and acceptable units can easily be extended and those herein were chosen simply to limit the development of *MassWateR* to a manageable scope.

## 3.2 Outliers

Checking the results file for outliers is an additional and essential step of QC screening. Outliers can have many causes, including data entry errors, laboratory processing errors, or anomalous environmental events. Identifying and potentially removing or correcting outliers is a critical part of quality control. The functions in *MassWateR* simply identify potential outliers to provide an opportunity for users to address these values. The decision on how to address

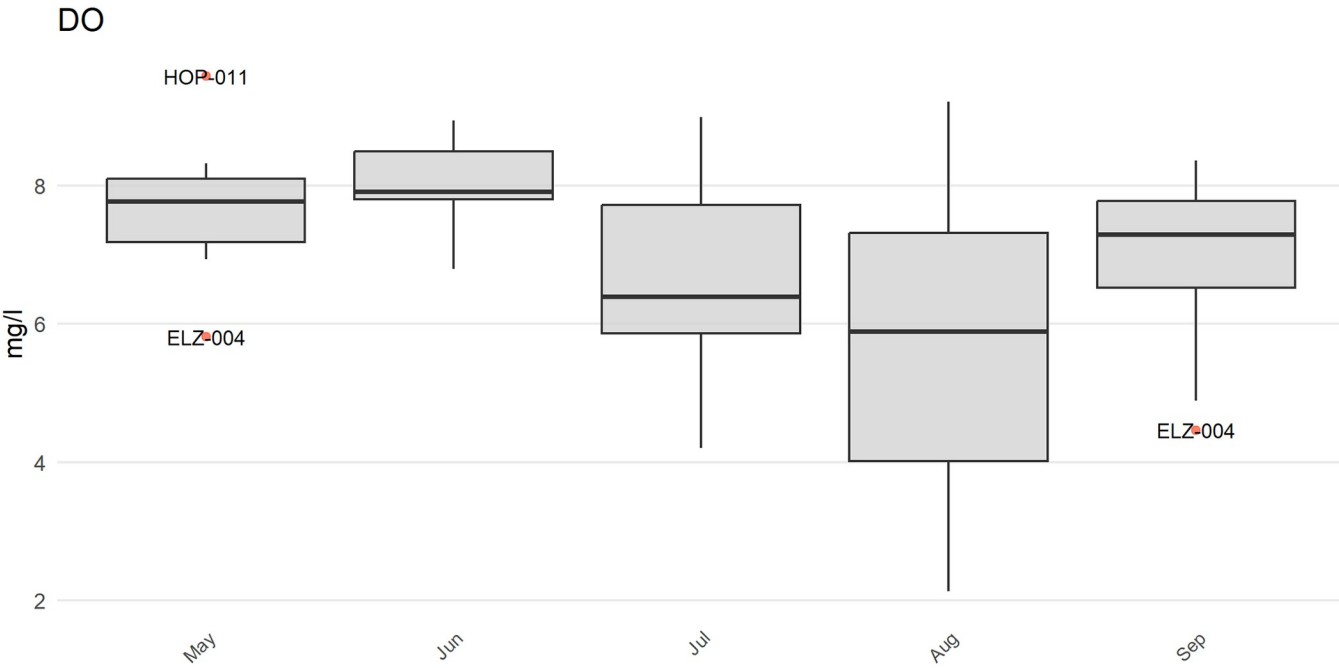

**Fig 3. Example plot showing identification of outliers in a water quality dataset.** The plot was created using the anlzMWRoutlier() function.

outliers remains with the user and no automated tools are provided in the package to correct or remove these values.

The anlzMWRoutlier() function uses the results file and data quality objectives for accuracy to plot potential outliers for a parameter by month. The accuracy file is used to automatically identify the y-axis scaling (as arithmetic or log) and to replace concentrations with appropriate values for those labelled as beyond detection. Outliers are defined using the standard definition of 1.5 times the interquartile range (the 25th to 75th percentile) of a parameter and can be visually identified as points above or below the whiskers in the boxplots. The station name for a point is also shown for identification. The param argument specifies the water quality parameter to evaluate and the group argument specifies how the boxplots are grouped (by year, month, week, or station). In the following example (Fig 3), three stations are identified as potential outliers for the respective month. Note that the outliers may differ slightly by the grouping choice as they are specific to each value on the x-axis.

anlzMWRoutlier(fset = fsetls, param = "DO", group = "month")

The outliers in the above plot can also be viewed as tabular output using outliers = TRUE to aid in their identification.

```
anlzMWRoutlier(fset = fsetls, param = "DO", group = "month",
outliers = TRUE)
# A tibble: 3 × 6
`Monitoring Location ID``Activity Start Date``Activity Start Time`
<chr> <dttm> <chr>
1 ELZ-004 2022-05-15 00:00:00 06:50
2 HOP-011 2022-05-15 00:00:00 06:55
3 ELZ-004 2022-09-11 00:00:00 07:20
# i 3 more variables: `Characteristic Name`<chr>, `Result
Value`<dbl>,
# `Result Unit`<chr>
```

A user can also jointly evaluate outliers for every water quality parameter in the results file. The anlzMWRoutlierall() function creates a Word document with images of boxplots for every parameter. Images for each parameter can also be created as standalone files. Here, the param argument does not need to be specified because all parameters are identified in the results file and processed accordingly. The following shows how to use the function to create a single Word file or individual image files in the working directory.

```
# create word output
anlzMWRoutlierall(fset = fsetls, group = 'month', format = 'word',
output_dir = getwd())
# create png output
anlzMWRoutlierall(fset = fsetls, group = 'month', format = 'png',
output_dir = getwd())
```

As in Fig 2, a user can identify outliers from the results, modify the file in Excel, and import the file again for further QC reporting or analysis.

## 3.3 Quality control reporting

The quality control functions in *MassWateR* are designed to create a single report that compares the water quality data in the results file (resdat) to data quality objectives in the accuracy (accdat, see the S1 File for the sample file included with the package) and frequency and completeness (frecomdat, see the S1 File for the sample file included with the package) files. In general, the QC checks for accuracy evaluate if laboratory and field duplicates, blanks, or spikes are within acceptable ranges. The QC checks for frequency and completeness can be used to evaluate if a sufficient number of records in the results file satisfy the accuracy checks and that sufficient QC data have been collected. The values in the accdat and frecomdat inputs are collectively described as data quality objectives (DQOs), such that the QC samples in resdat must satisfy these objectives to be considered accurate and precise data for use in regulatory or other assessments by water quality professionals.

The following describes the types of QC measurements that are evaluated by *MassWateR*:

- *Field Blanks*: Measurements in the field to verify that the parameter is below a certain threshold, e.g., below the detection limit of the field equipment

- *Lab Blanks*: Laboratory samples to verify that the parameter is below a certain threshold, e.g., below the detection limit of the laboratory analysis method

- *Field Duplicates*: Duplicate measurements in the field to assess the similarity of values, i.e., precision is high

- *Lab Duplicates*: Duplicate laboratory samples to assess the similarity of values, i.e., precision is high

- *Calibration Checks*: Measurements in the field to verify an instrument returns a known quantity of a parameter, i.e., to verify accuracy of field sampling equipment

- *Lab Spikes*: Laboratory samples to verify if the processing equipment returns a known quantity of a parameter, i.e., to verify accuracy of laboratory equipment

Additionally, *MassWateR* allows a user to include "qualified samples" in the results file. These samples are similar as those from the above criteria and are identified by the user as having special circumstances. For example, field duplicates are typically in pairs and the second measurement can be entered as a qualified sample in the same row of the first sample under

**QC Review**

*Organization Name*

Jul 06, 2023
Prepared by: ___________________
QAPP version: ___________________

**Data Quality Objectives**

| Parameter | Field Duplicate | Lab Duplicate | Field Blank | Lab Blank | Spike/Check Accuracy | % Completeness |
|---|---|---|---|---|---|---|
| | | | Frequency % | | | |
| Ammonia | 10 | 5 | 10 | 5 | 5 | 90 |
| DO | 10 | - | - | - | - | 90 |
| E.coli | 10 | 5 | 10 | 5 | - | 90 |
| Nitrate | 10 | 5 | 10 | 5 | 5 | 90 |
| Sp Conductance | 10 | 10 | - | 10 | 10 | 90 |
| TP | 10 | 5 | 10 | 5 | 5 | 90 |
| Water Temp | 10 | 10 | - | - | 10 | 90 |
| pH | 10 | 10 | - | - | 10 | 90 |

| Parameter | uom | MDL | UQL | Value Range | Field Duplicate | Lab Duplicate | Field Blank | Lab Blank | Spike/Check Accuracy |
|---|---|---|---|---|---|---|---|---|---|
| Ammonia | mg/l | 0.1 | - | all | < 30% | < 20% | BDL | BDL | <= 15% |
| DO | mg/l | - | - | < 4 | < 20% | - | - | - | - |
| DO | mg/l | - | - | >= 4 | < 10% | - | - | - | - |
| E.coli | MPN/100ml | 1 | - | <50 | < log30% | < log30% | BDL | BDL | - |
| E.coli | MPN/100ml | 1 | - | >=50 | < log20% | < log20% | BDL | BDL | - |
| Nitrate | mg/l | 0.05 | - | all | < 30% | < 20% | BDL | BDL | <= 15% |
| Sp Conductance | uS/cm | - | - | < 250 | < 30% | < 30% | - | <= 50 | <= 50 |
| Sp Conductance | uS/cm | - | 10000 | >= 250 | < 20% | < 20% | - | <= 50 | <= 50 |
| TP | mg/l | 0.01 | - | < 0.05 | <= 0.01 | <= 0.01 | BDL | BDL | <= 0.01 |
| TP | mg/l | 0.01 | - | >= 0.05 | < 30% | < 20% | BDL | BDL | <= 15% |
| Water Temp | deg C | - | - | all | <= 1.0 | <= 1.0 | - | - | <= 1.0 |
| pH | - | - | - | all | <= 0.5 | <= 0.5 | - | - | <= 0.2 |

Notes:

| QC Review | Page 1 of 16 | 7/6/2023 |
|---|---|---|

**QC Frequencies for 5/15/2022 to 9/11/2022**

| Parameter | Field Duplicate | Lab Duplicate | Field Blank | Lab Blank | Spike/Check Accuracy |
|---|---|---|---|---|---|
| Ammonia | 9% | 23% | 16% | 16% | 21% |
| DO | 22% | - | - | - | - |
| E.coli | 17% | 33% | 33% | 0% | - |
| Nitrate | 10% | 50% | 35% | 25% | 50% |
| pH | 22% | 35% | - | - | 41% |
| Sp Conductance | 22% | 35% | - | 43% | 43% |
| TP | 10% | 33% | 23% | 10% | 31% |
| Water Temp | 22% | 35% | - | - | 39% |

| Type | Parameter | Number of Data Records | Number of Dups/Blanks/Spikes | Frequency % | Hit/Miss |
|---|---|---|---|---|---|
| Field Duplicates | | | | | |
| | Ammonia | 43 | 4 | 9% | MISS |
| | DO | 49 | 11 | 22% | |
| | E.coli | 12 | 2 | 17% | |
| | Nitrate | 20 | 2 | 10% | |
| | Sp Conductance | 49 | 11 | 22% | |
| | TP | 48 | 5 | 10% | |
| | Water Temp | 49 | 11 | 22% | |
| | pH | 49 | 11 | 22% | |
| Lab Duplicates | | | | | |
| | Ammonia | 43 | 10 | 23% | |
| | E.coli | 12 | 4 | 33% | |
| | Nitrate | 20 | 10 | 50% | |
| | Sp Conductance | 49 | 17 | 35% | |
| | TP | 48 | 16 | 33% | |
| | Water Temp | 49 | 17 | 35% | |
| | pH | 49 | 17 | 35% | |
| Field Blanks | | | | | |
| | Ammonia | 43 | 7 | 16% | |
| | E.coli | 12 | 4 | 33% | |
| | Nitrate | 20 | 7 | 35% | |
| | TP | 48 | 11 | 23% | |
| Lab Blanks | | | | | |
| | Ammonia | 43 | 7 | 16% | |
| | E.coli | 12 | 0 | 0% | MISS |
| | Nitrate | 20 | 5 | 25% | |
| | Sp Conductance | 49 | 21 | 43% | |
| | TP | 48 | 5 | 10% | |
| Lab Spikes / Instrument Checks | | | | | |

| QC Review | Page 2 of 16 | 7/6/2023 |
|---|---|---|

**Fig 4. The first two of sixteen pages of the quality control report created by qcMWRreview() that evaluates the results data relative to data quality objectives.** The first page shows the data quality objectives for accuracy, frequency, and completeness. The second page shows QC results for frequency and completeness. Parameters shown in red or marked as 'MISS' failed the data quality objectives. Users can edit the Word file as needed, e.g., entering the organization name or adding notes.

the "QC Reference Value" column of the results file. This eliminates row repetition and facilitates preparation of the results file for the user.

The qcMWRreview() function creates a single QC report as a Word document that evaluates all data in the results file using the DQOs in the accdat and frecomdat input files. This file includes several tables created by individual *MassWateR* functions, where each describes relevant evaluations for the QC assessment. The file is created as follows, which typically requires less than a minute to complete and is followed by a message in the console indicating the report was successfully created and the path where the Word file is located. A user can then further edit the Word document as needed. The first two pages of the QC report are shown in Fig 4.

```
qcMWRreview(fset = fsetls, output_dir = getwd())
#> Report created successfully! File located at /tmp/RtmpUzzrbC/qcreview.docx
```

The QC report is built using several functions that can be used individually as needed. In particular, the tabMWRacc(), tabMWRfre(), and tabMWRcom() create *flextable* [24] objects that can be viewed in RStudio and are compatible with Word output. These functions are useful for understanding how the QC checks are created for the separate components of the QC report. For example, the tabMWRacc() function evaluates accuracy checks for field duplicates, lab duplicates, field blanks, lab blanks, and lab spikes/instrument checks for QC records in the

**Table 2. Summary of quality control results for the accuracy data quality objectives.** The number of checks and failed checks (misses) for field duplicates, lab duplicates, field blank, lab blanks, and lab spikes / instruments are shown for each parameter. Only ammonia and total phosphorus are shown for brevity.

| Type | Parameter | Number of QC Checks | Number of Misses | % Acceptance |
|---|---|---|---|---|
| Field Duplicates | | | | |
| | Ammonia | 4 | 1 | 75% |
| | TP | 5 | 1 | 80% |
| Lab Duplicates | | | | |
| | Ammonia | 10 | 0 | 100% |
| | TP | 16 | 0 | 100% |
| Field Blanks | | | | |
| | Ammonia | 7 | 0 | 100% |
| | TP | 11 | 1 | 91% |
| Lab Blanks | | | | |
| | Ammonia | 7 | 1 | 86% |
| | TP | 5 | 0 | 100% |
| Lab Spikes / Instrument Checks | | | | |
| | Ammonia | 9 | 0 | 100% |
| | TP | 15 | 0 | 100% |

results file based on DQOs in the accuracy file (Table 2). The function can return a summary of all checks as follows (only ammonia and total phosphorus are shown for brevity):

```
tabMWRacc(fset = fsetls, type = "summary")
```

The table shows the types and amounts of QC checks applied to each parameter and which of those checks did not satisfy the DQOs in the accuracy file. For example, there were four field duplicate records for ammonia in the results file and only one of those records included a duplicate value outside of the acceptable range, i.e., there was a 75% acceptance rate for the four field duplicate records. This function can also return a matrix of only acceptance rates for a more compact view of the QC accuracy (Table 3).

```
tabMWRacc(fset = fsetls, type = "percent")
```

The cells in red in Table 3 show that four of the QC checks for three of the parameters did not meet the accuracy DQOs. All other parameters and checks in green had enough QC checks to satisfy the DQOs (note that the table colors are color-blind friendly). Empty cells include checks where no QC records were available in the results file, e.g., no lab duplicates for dissolved oxygen. The empty cells are typically for QC checks that do not readily apply to a

**Table 3. Condensed summary of quality control results for the accuracy data quality objectives.** The table shows only the percentages from Table 2, with cells colored based on whether the required number of checks were met.

| Parameter | Field Duplicate | Lab Duplicate | Field Blank | Lab Blank | Spike/Check Accuracy |
|---|---|---|---|---|---|
| Ammonia | 75% | 100% | 100% | 86% | 100% |
| DO | 100% | - | - | - | - |
| E.coli | 100% | 100% | 100% | - | - |
| Nitrate | 100% | 100% | 100% | 100% | 90% |
| pH | 100% | 94% | - | - | 95% |
| Sp Conductance | 100% | 100% | - | 95% | 100% |
| TP | 80% | 100% | 91% | 100% | 100% |
| Water Temp | 100% | 100% | - | - | 95% |

**Table 4. Detailed information on quality control results for field blanks.** Results show the individual dates for the quality control checks, the results, and the data quality threshold for comparison. Only ammonia and total phosphorus are shown for brevity.

| Parameter | Date | Site | Result | Threshold | Hit/Miss |
|---|---|---|---|---|---|
| Ammonia | | | | | |
| | 2022-05-15 | | BDL | 0.1 mg/l | |
| | 2022-06-12 | | BDL | 0.1 mg/l | |
| | 2022-07-17 | | BDL | 0.1 mg/l | |
| | 2022-07-17 | | BDL | 0.1 mg/l | |
| | 2022-08-14 | | BDL | 0.1 mg/l | |
| | 2022-08-14 | | BDL | 0.1 mg/l | |
| | 2022-09-11 | | BDL | 0.1 mg/l | |
| TP | | | | | |
| | 2022-05-15 | | BDL | 0.01 mg/l | |
| | 2022-05-15 | | BDL | 0.01 mg/l | |
| | 2022-06-12 | | BDL | 0.01 mg/l | |
| | 2022-06-12 | | BDL | 0.01 mg/l | |
| | 2022-07-17 | | BDL | 0.01 mg/l | |
| | 2022-07-17 | | BDL | 0.01 mg/l | |
| | 2022-07-17 | | 0.01 mg/l | 0.01 mg/l | MISS |
| | 2022-08-14 | | BDL | 0.01 mg/l | |
| | 2022-08-14 | | BDL | 0.01 mg/l | |
| | 2022-09-11 | | BDL | 0.01 mg/l | |
| | 2022-09-11 | | BDL | 0.01 mg/l | |

The example shows that all but one field blank passed the checks, where most were below the detection limit (BDL) of the laboratory equipment. Only one sample taken on July 17th for total phosphorus (TP) was at or above the threshold (i.e., was a "MISS" for the DQO target).

parameter. For example, dissolved oxygen is measured in the field with monitoring equipment, such that lab QC checks are not relevant.

Detailed information about the QC checks for an individual parameter can be obtained by changing the inputs to the type and accchk arguments. For example, the results for every field blank check and every parameter can be obtained using type = "individual" and accchk = "Field Blanks" (Table 4). This is the same information that is summarized using type = "summary" or type = "percent". Only ammonia and total phosphorus are shown for brevity.

```
tabMWRacc(fset = fsetls, type = "individual", accchk = "Field Blanks")
```

Additional functions for creating QC tables include tabMWRfre() for assessing if QC samples were conducted at the targeted frequency and tabMWRcom() for assessing completeness checks that compare the number of regular samples (field measurements or lab samples) to the number of qualified samples, i.e., a measure of the proportion of the data that is usable. As with all the QC functions, these can be used individually within an R session to quickly view QC results or the qcMWRreview() function can be used to create a complete report that combines outputs from the individual functions. The summarized report can then be submitted to an appropriate regulatory agency for review to ensure that any submitted datasets fulfill appropriate data quality objectives.

### 3.4 Analysis

The analysis functions in *MassWateR* provide a streamlined approach to quickly evaluate data in the results file. Although several base R functions and supporting packages can be used to

develop custom assessments, it was recognized that many users will not be comfortable developing their own visualization routines. As such, the analysis functions were designed for a rapid overview of the results that require minimal decisions by the user to create the output. Several default settings described below allow further customization of the output as desired by the user. Additionally, all plots returned by *MassWateR* are ggplot class objects and can be modified using conventional *ggplot2* functions (see the website vignette). The four primary analysis functions can be used to analyze seasonal trends, trends by date, data by site, and spatially using maps.

Seasonal trends can be evaluated using the anlzMWRseason() function that summarizes results for a single parameter using boxplots or barplots with seasonal groups assigned to months or weeks of the year (Fig 5). Boxplots or barplots can also include jittered points of the observations on top or only the jittered points can be shown. Fig 5 demonstrates a jittered boxplot of dissolved oxygen observations by month.

```
anlzMWRseason(fset = fsetls, param = "DO", thresh = "fresh",
group = "month", type = "jitterbox")
```

The arguments specify different options available for the plot, including which parameter to plot in the results file (param), the type of threshold line(s) to show (thresh), the grouping (group), and the type of plot (type as jittered points over boxplots). The thresh argument, in particular, is used to plot relevant thresholds of interest based on the parameter and if the samples are from freshwater or marine sites. The thresholdMWR data object included with the package includes thresholds for ten common parameters that are either regulatory standards or recommended safe limits for different designated uses in freshwater or marine environments. Each threshold includes a source that is displayed in the legend above the plot. In the above example, the threshold line applies to class A or B freshwater environments that support

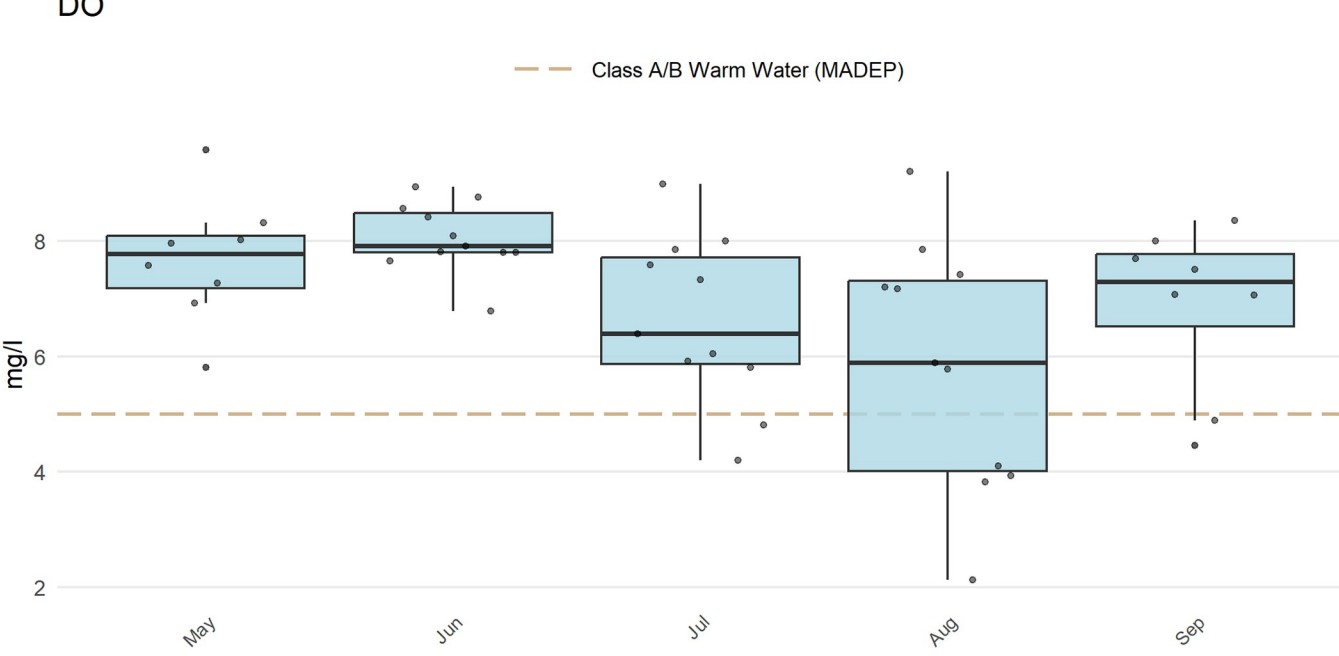

**Fig 5. Example plot showing water quality results grouped by month.** The plot was created using the anlzMWRseason() function.

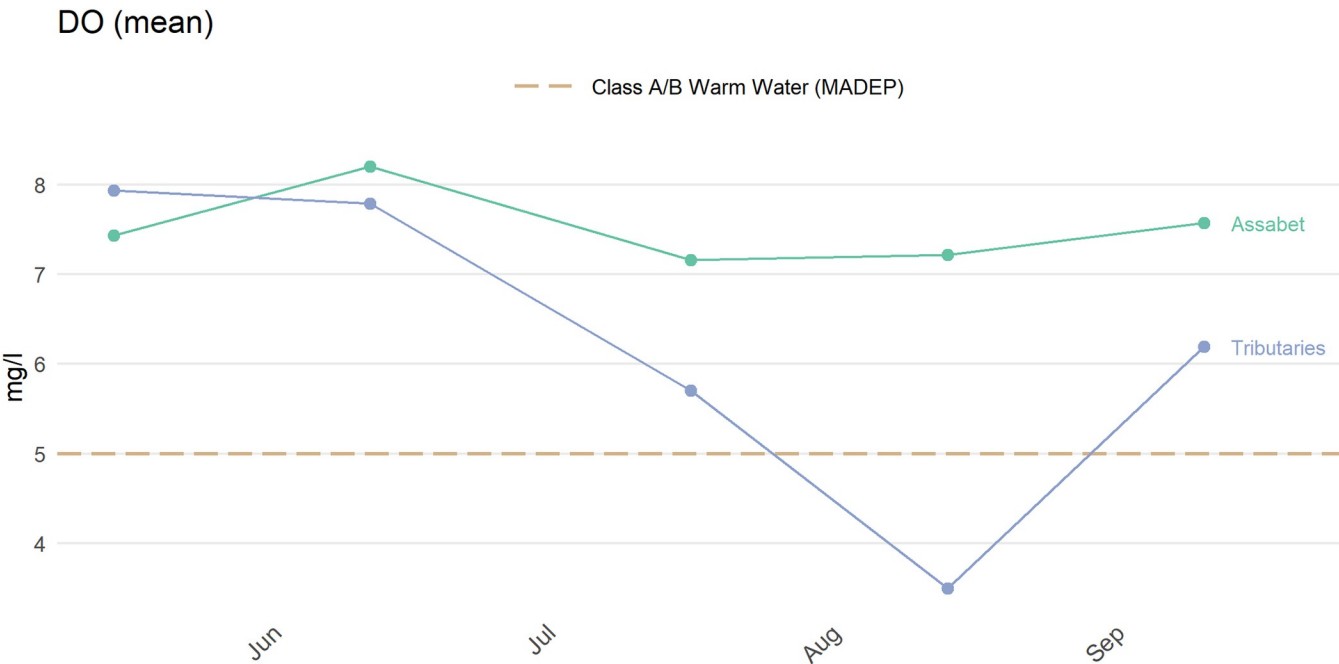

**Fig 6. Example plot showing water quality results grouped by sample date.** The plot was created using the anlzMWRdate() function.

warmwater fisheries, as defined by MADEP. Threshold lines can be suppressed (thresh = "none") or user-specific thresholds can be added using additional *ggplot2* functionality (i.e., with geom_hline() added to the plot output).

The y-axis scaling of the plot is also determined automatically from the data inputs. The DQO file for accuracy includes information on the distribution of each parameter, i.e., parameters with "log" in any of the columns are plotted on log10-scale, otherwise arithmetic. This behavior is controlled by the yscl argument, where the default is "auto" which indicates information on scaling is obtained automatically from the DQO file. Setting yscl = "linear" or yscl = "log" will set the axis as linear or log10-scale, respectively, regardless of the information in the DQO file.

The anlzMWRdate() function plots results as time series on the date the samples or measurements were collected (Fig 6). This information can be used to evaluate how samples have changed at individual sites over time or as aggregate samples across sites. In the former case, individual points and lines are used for each site with appropriate labels. For the latter case, sites are aggregated and summary statistics are shown as the mean with optional 95% confidence intervals for each date. All sites can be aggregated by sample dates or by "location groups" in the site metadata file. For example, all sites along the same river or tributary can be aggregated. Fig 6 shows an aggregation of sites by the mainstem of the Assabet river and its tributaries using the group = "locgroup" argument. A user can specify any desired grouping in the Location Group column of the site metadata file.

```
anlzMWRdate(fset = fsetls, param = 'DO', group = 'locgroup',
thresh = 'fresh')
```

The anlzMWRsite() function can be used to view data organized by site (Fig 7). This function summarizes results for a single parameter using boxplots or barplots separately for each site on the x-axis. Boxplots or barplots can also include overlaid jittered points of the observations or only the jittered points can be shown, as for the anlzMWRseason() function. Results at

## DO, data filtered by sites, result attributes

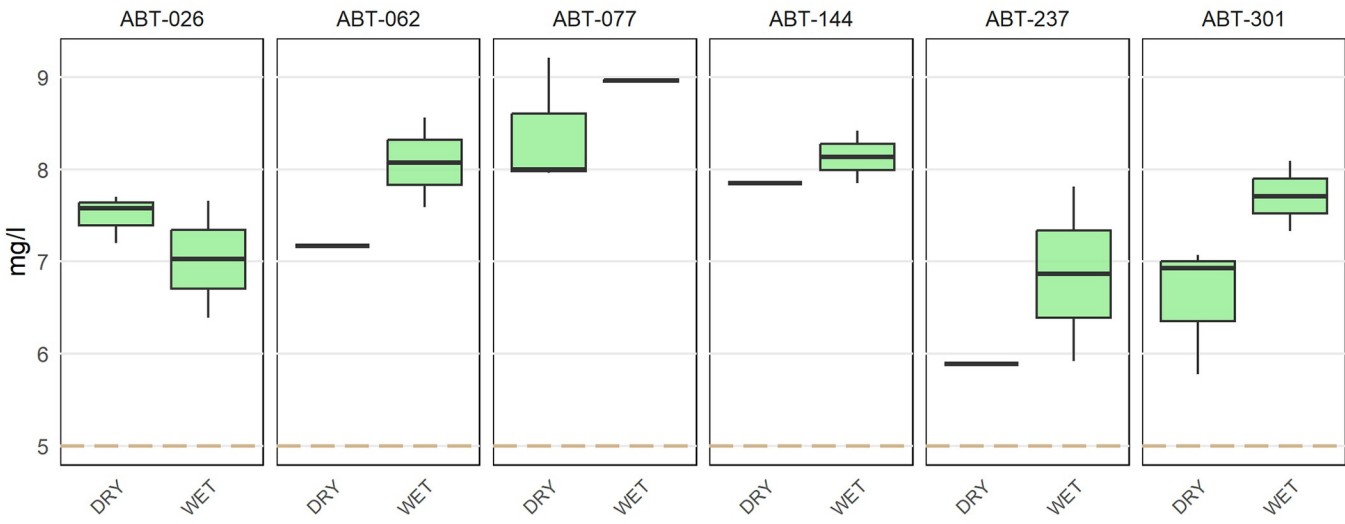

**Fig 7. Example plot showing water quality results grouped by site.** The plot was created using the anlzMWRsite() function.

each site can also be grouped by the Result Attribute column in the results file, where a user can enter arbitrary grouping criteria for samples. For example, sites can be grouped by wet or dry conditions if this information is included in the results file to evaluate the effects of precipitation on water chemistry or pollutants. Individual sites to plot can also be specified with the site argument, where the default is to plot all sites.

```
anlzMWRsite(fset = fsetls, param = "DO", thresh = "fresh", type =
"box",
site = c("ABT-026", "ABT-062", "ABT-077", "ABT-144", "ABT-237", "ABT-
301"),
resultatt = c('DRY', 'WET'), byresultatt = TRUE)
```

Finally, maps of results at each site can be created with the anlzMWRmap() function (Fig 8). Parameter values at each site are summarized with colors indicating relative values and the maps can include various spatial information for context. For example, hydrologic lines and waterbodies from the National Hydrograpy Dataset (NHD) can be shown with varying level of detail and base maps can be included from the *ggmap* package [25]. NHD data were included to provide more specific information on hydrologic features of interest because of insufficient detail provided by standard basemaps. These datasets are available from an external source and clipped to an approximate bounding box for the selected stations. Currently, only flowlines and waterbodies that intersect Massachusetts are included, although examples are provided in a package vignette that demonstrate how to include custom shapefiles in the map. Fig 8 shows mean dissolved oxygen concentrations using the medium level of detail for the NHD maps (addwater = "medium").

```
anlzMWRmap(fset = fsetls, param = "DO", addwater = "medium")
```

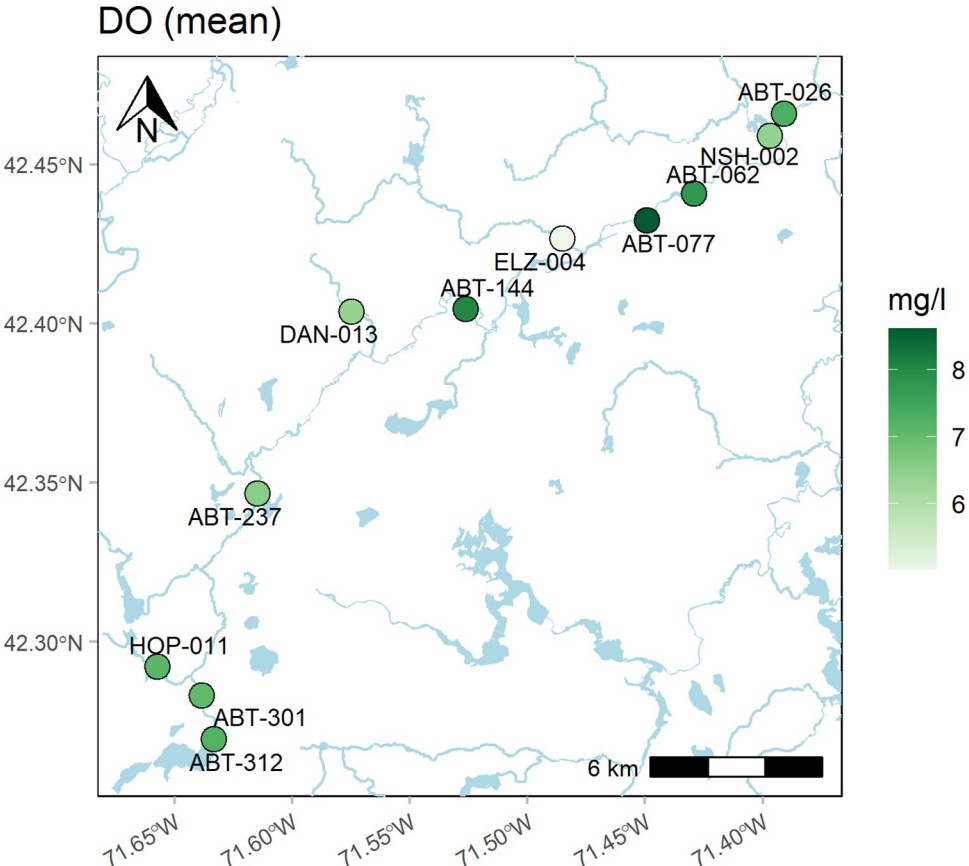

**Fig 8. Example map showing water quality results averaged by sampling location.** The map was created using the anlzMWRmap() function.

Samples and measurements across dates in the map can be aggregated differently using the sumfun argument. If sumfun = "auto" (default), the mean is used where the distribution is determined automatically from the DQO file for accuracy, i.e., parameters with "log" in any of the columns are summarized with the geometric mean, otherwise arithmetic. Additional valid R summary function will be applied if passed to sumfun ("mean", "geomean", "median", "min", "max"). This argument also applies to other analysis functions where the data can be aggregated across dates or locations.

### 3.5 Data submission

The last part of the *MassWateR* workflow is preparation of data for submission to WQX. A single function, tabMWRwqx(), is provided to format data inputs using a template designed for data submission. This function will export a single Excel workbook with three sheets, named "Project", "Locations", and "Results", all of which are required by USEPA for data submission. The output is populated with as much content as possible based on information in the input files and the user can manually adjust the workbook prior to submission. All required columns are present, but individual rows must be verified for completeness and accuracy by the user before uploading the data.

The workbook can be created as follows by including the required files and specifying an output directory where the Excel file is saved. Once the function is done running, a message

indicating success and where the file is located is returned. Submitting the data to WQX simply requires uploading the output file into the data portal. After passing checks within WQX, the data are archived in the portal within 3–5 business days and made accessible for public use.

```
tabMWRwqx(fset = fsetls, output_dir = getwd())
#> Excel workbook created successfully! File located at /tmp/
RtmpBrk9nw/wqxtab.xlsx
```

Additional templates and instructions for data submission are available on the package website. It is assumed that users are familiar with the WQX data submission portal if they are already using the package, though training materials from USEPA are available. *MassWateR* eliminates the need to format the data by hand and it is expected to increase the amount of data made available on the WQP (via WQX upload) as the user base for the package increases. Further, the authors worked with USEPA WQX staff to improve WQX handling of Quality Control-coded data, by improving the list of potential field names and highlighting the need to better connect paired data (e.g., activity types "Quality Control Lab Duplicate 1" and "Quality Control Lab Duplicate 2" were created to pair lab duplicates). Custom WQX import configurations were also developed to streamline data submission, which were made available publicly to all WQX users.

## 4. Discussion

To ensure that *MassWateR* is a known resource for potential users and to encourage its use for QC reporting, analysis, and data submission, a community of practice [e.g., 26] was established during package development and following its initial release on CRAN (January, 2023). This included several beta testing and training workshops to gather feedback on anticipated data analysis workflows and to educate potential users on appropriate use of the package [27,28]. Many users have not previously been exposed to R for data analysis and a substantial portion of the trainings included an introduction to R, as well as *MassWateR*. Emphasis was placed on simple use of the core functions, as opposed to developing custom workflows that combined core R functions with *MassWateR*. However, users were encouraged to learn how to extend the use of the package by leveraging additional R packages, such as *ggplot2* [17] to create additional visualizations for modifying the existing analysis plots in *MassWateR*. For example, a vignette was included on the package website to demonstrate how these plots can be modified for custom output using *ggplot2*. Further, a community forum was created as a resource for users to post questions about the package and for others to view the discussion if similar issues were encountered. This approach followed the model used by other popular web forums (e.g., StackOverflow) for troubleshooting software issues by minimizing duplication of topics through sharing solutions in a public forum.

As noted above, *MassWateR* was developed to meet specific needs of water quality professionals in Massachusetts, but the principles for QC reporting, analysis, and WQX data submission are largely universal and the package can be used outside of the state as long as the following minor limitations are addressed. First, some of the parameter thresholds that are visually overlaid on the analysis plots are unique to Massachusetts, whereas others apply more broadly, such as those defined using standards from the USEPA. Users can simply omit the thresholds if they do not apply using thresh = "none" or add custom thresholds using standard *ggplot2* functions. Second, the NHD waterbodies used by anlzMWRmap() are specific to watersheds that intersect Massachusetts. Users can omit these layers from the plot using addwater = NULL, add a universal base map using the maptype argument, or add custom waterbody layers as simple features objects [29] using the geom_sf() function from *ggplot2*.

Third, the QC report created by qcMWRreview() uses a format vetted by MADEP. This format may not meet the requirements of other state organizations, although the reporting principles are generally universal and the resulting Word document can easily be manually modified. Finally, minor components of the Excel file for WQX submission created by tabMWRwqx() are specific to Massachusetts. This includes the timezone for the WQX Activity Start Time Zone field and some default entries for sample collection methods that include "MassWateR" in the text. Each of these can also be modified by hand in the output file. Future enhancements and additions to *MassWateR* will likely include automated tools to make the package more applicable outside of Massachusetts.

Additional future work to improve the functionality of *MassWateR* is also expected as the user base increases and the functions mature with additional bug fixes or minor enhancements. Specifically, the inclusion of historical data for some of the analysis functions could provide additional context on status and trends for monitoring data at specific locations. This enhancement would require the extraction of existing data included in the WQP, which would not be significantly challenging given the robust web retrieval tools already available. Existing R packages leverage these tools [e.g., *dataRetrieval*, 19] and a similar approach could be used by *MassWateR*. A second and more challenging enhancement would be the ability to work with continuous monitoring data collected at high temporal resolution with equipment deployed *in situ* [e.g., 10]. These data present additional challenges not encountered with routine samples collected at longer time intervals, including increased data volume and additional QC needs [10,30]. For the latter, automated tools are needed for detecting and handling QC issues common with monitoring equipment deployed in the field for long duration, such as instrument drift, biofouling, or missing data. More complex methods for detecting outliers in continuous monitoring data beyond the existing tools in *MassWateR* will also be needed [31].

## 5. Conclusions

The *MassWateR* package represents an important set of functions that are expected to significantly improve how monitoring programs and resource managers implement QC assessments and apply exploratory analyses to water quality monitoring data. The package can also expedite preparation of data for submission to the largest water quality database in the United States, which will likely contribute to the new data made available through the WQP. These activities are critically needed to ensure that monitoring data are of sufficient quality and quantity for use in regulatory applications or routine assessments of status and trends of environmental resources. As mentioned throughout this paper, *MassWateR* was developed to address specific needs for resource management professionals in Massachusetts, although the workflow in Fig 1 can easily be applied to data collected elsewhere. Future development of the package will not only make the package functions more generalizable to other locations, but also provide additional features for working with continuous and historical monitoring data. The community of practice developed for *MassWateR* is expected to grow and the package will be supported by the authors as the user base increases.

## Supporting information

**S1 File. *MassWateR* parameters and example data quality objective inputs.**
(DOCX)

## Acknowledgments

We thank the early users of *MassWateR* that contributed ideas on improving the package during initial testing. We are grateful for discussions with the Massachusetts Department of Environmental Protection that guided the presentation of results in the quality control assessments

and WQX staff at USEPA for consult on submitting data formatted with *MassWateR*. We also thank two reviewers and the academic editor of the journal for helpful comments that improved the manuscript.

## Author Contributions

**Conceptualization:** Marcus W. Beck, Benjamen Wetherill, Jillian Carr.

**Data curation:** Marcus W. Beck, Benjamen Wetherill, Jillian Carr.

**Formal analysis:** Marcus W. Beck, Benjamen Wetherill, Jillian Carr.

**Funding acquisition:** Jillian Carr.

**Investigation:** Marcus W. Beck, Benjamen Wetherill, Jillian Carr.

**Methodology:** Marcus W. Beck, Benjamen Wetherill, Jillian Carr.

**Project administration:** Jillian Carr.

**Software:** Marcus W. Beck, Benjamen Wetherill.

**Supervision:** Jillian Carr.

**Validation:** Benjamen Wetherill.

**Visualization:** Marcus W. Beck, Benjamen Wetherill, Jillian Carr.

**Writing – original draft:** Marcus W. Beck.

**Writing – review & editing:** Marcus W. Beck, Benjamen Wetherill, Jillian Carr.

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
