## [Decision Letter · Decision Letter 0]

6 Sep 2023

PONE-D-23-27749MassWateR: Improving quality control, analysis, and sharing of water quality data

PLOS ONE

Dear Dr. Beck,

Thank you for submitting your manuscript to PLOS ONE. After careful consideration, we feel that it has merit but does not fully meet PLOS ONE’s publication criteria as it currently stands. Therefore, we invite you to submit a revised version of the manuscript that addresses the points raised during the review process.

ACADEMIC EDITOR

Though one of the reviewers recommended a complete rejection, I found that this study could be of benefit to the community especially in the industry 4.0 era. However, I request that you take into consideration the following while revising your manuscript.1.
You should describe explicitly how the package provides a better alternative to all the other existing packages. Where applicable, cite the weaknesses of the existing packages with supporting citations. 2.
Make appropriate citations to literature sources. For example, in L26 and L29, you talk about an Act and an EU framework whose sources are not cited. The same applies to L39-40, where you argue out why this communication could be important but without any supporting citation(s).  Similar omissions are evident in the DISCUSSION where you should make appropriate citation of the ‘‘community of practice established’’ and any outcomes (L521-537, and so forth). 3.
L87-99 appears to should have been part of the introduction, highlighting what the MassWateR brings onboard. 4.
Figure 1 could best be presented as a flow/block diagram.5.
For Table 4, it would be important to provide all the parameters tested as a supplementary file to support the submission. Please ensure that the full details of the algorithms designed are provided, and where there are restrictions, it should otherwise be stated.6. To improve the chances of manuscript suitability for PLOS ONE, ensure that the criteria listed for manuscript types (https://journals.plos.org/plosone/s/submission-guidelines#loc-methods-software-databases-and-tools) are met. i.e. Utility, Validation and Availability for articles describing new/improved METHODS, SOFTWARE, DATABASES, and TOOLS. Please submit your revised manuscript by Oct 21 2023 11:59PM. If you will need more time than this to complete your revisions, please reply to this message or contact the journal office at plosone@plos.org. Please include the following items when submitting your revised manuscript:A rebuttal letter that responds to each point raised by the academic editor and reviewer(s). You should upload this letter as a separate file labeled 'Response to Reviewers'.A marked-up copy of your manuscript that highlights changes made to the original version. You should upload this as a separate file labeled 'Revised Manuscript with Track Changes'.An unmarked version of your revised paper without tracked changes. You should upload this as a separate file labeled 'Manuscript'.

We look forward to receiving your revised manuscript.

Kind regards,

Timothy Omara, PhD

Academic Editor

PLOS ONE

Journal Requirements:

Reviewers' comments:

Reviewer's Responses to Questions

**Comments to the Author**

1. Is the manuscript technically sound, and do the data support the conclusions?

Reviewer #1: Yes

Reviewer #2: No

2. Has the statistical analysis been performed appropriately and rigorously? 

Reviewer #1: N/A

Reviewer #2: Yes

3. Have the authors made all data underlying the findings in their manuscript fully available?

Reviewer #1: Yes

Reviewer #2: Yes

4. Is the manuscript presented in an intelligible fashion and written in standard English?

Reviewer #1: Yes

Reviewer #2: Yes

5. Review Comments to the Author

Reviewer #1: The paper describes a new R package for the processing of water quality information, mainly following US standards, but could be adapted for other regions. It is well written and easy to follow. I have only a few minor comments. Several of the figures were missing from the submission, so I could not check them (Figure 3, and Figure 5 following).

L27 and 29 please cite respective regulatory frameworks

L108 should be „their data in several ways“

L109 should be „outlier checks“

L137 should be function - spelling error

L327 should be „to evaluate“

L334-335 I guess more correct would be to state „to verify that the parameter is below a certain threshold, e.g. below detection limit“. Even if the parameter has a value of zero - the parameter itself is not absent

336-337 same

338-339 More correct would be „to assess the similarity of values, i.e. precision is high. (Precision is a value that is calculated from bot measurements - do not understand how each measurement can have its own precision)

385 Add: „in red in Table 3 show…“

439 spelling „can be“

506 I guess public use is better than public consumption

Reviewer #2: The availability of reliable monitoring data is the basis of scientific research. This study collected discrete surface water quality data generated by MassWateR software package and analyzed the trend of parameters. The study has some fundamental significance. However, this paper is more like a technical manual than a scientific research paper. Therefore, I would recommend the authors to submit the paper to a technical journal rather than a research-oriented journal.

In addition, the transferability of the developed tool needs to be demonstrated and analyzed, which is directly related to its application value. The authors should give concrete examples in this area;

Finally, the quality of the authors' images is so poor that improving the clarity is very necessary.

6. PLOS authors have the option to publish the peer review history of their article (what does this mean?). If published, this will include your full peer review and any attached files.

Reviewer #1: No

Reviewer #2: No

---

## [Author Response · Author response to Decision Letter 0]

26 Sep 2023

Please view the response to reviewers included with our resubmission.

---

## [Decision Letter · Decision Letter 1]

5 Oct 2023

PONE-D-23-27749R1MassWateR: Improving quality control, analysis, and sharing of water quality dataPLOS ONE

Dear Dr. Beck,

Thank you for submitting your manuscript to PLOS ONE. After careful consideration, we feel that it has merit but does not fully meet PLOS ONE’s publication criteria as it currently stands. Therefore, we invite you to submit a revised version of the manuscript that addresses the points raised during the review process.

We look forward to receiving your revised manuscript.

Kind regards,

Timothy Omara, PhD

Academic Editor

PLOS ONE

Journal Requirements:

Reviewers' comments:

Reviewer's Responses to Questions

**Comments to the Author**

1. If the authors have adequately addressed your comments raised in a previous round of review and you feel that this manuscript is now acceptable for publication, you may indicate that here to bypass the “Comments to the Author” section, enter your conflict of interest statement in the “Confidential to Editor” section, and submit your "Accept" recommendation.

Reviewer #1: All comments have been addressed

Reviewer #2: All comments have been addressed

2. Is the manuscript technically sound, and do the data support the conclusions?

Reviewer #1: Yes

Reviewer #2: Yes

3. Has the statistical analysis been performed appropriately and rigorously? 

Reviewer #1: N/A

Reviewer #2: Yes

4. Have the authors made all data underlying the findings in their manuscript fully available?

Reviewer #1: Yes

Reviewer #2: Yes

5. Is the manuscript presented in an intelligible fashion and written in standard English?

Reviewer #1: Yes

Reviewer #2: Yes

6. Review Comments to the Author

Reviewer #1: The authors have successfully revised the manuscript and convincingly responded to all reviewer comments. I therefore recommend publication.

Reviewer #2: The authors have improved over the previous version, but there are still some issues that need to be revised, the main comments are as follows:

1. it is recommended to add information about the data of the mean values as well as information about the true values in Figures 3 and 6, so that the reader can understand more information;

2. the quality of Figure 4 is very low and it is not even possible to see any content information;

3. the dimensions of Figure 8 are recommended to use vertical arrangement; and the scale labeling form is not standardized; the DO color labeling colors of the sites are more distinguishable;

4. how do the predictions of water quality parameters in this study compare with other common models?

5. some recent deep learning models for water quality prediction should be mentioned, e.g., 10.1016/j.ejrh.2023.101331; 10.1007/s11783-023-1688-y; 10.1016/j.ecolind.2023.109882; 10.1016/j.jhydrol.2023. 129649.

7. PLOS authors have the option to publish the peer review history of their article (what does this mean?). If published, this will include your full peer review and any attached files.

Reviewer #1: No

Reviewer #2: No

---

## [Author Response · Author response to Decision Letter 1]

8 Oct 2023

Please see the response to reviewers attachment included in our resubmission.

---

## [Decision Letter · Decision Letter 2]

19 Oct 2023

MassWateR: Improving quality control, analysis, and sharing of water quality data

PONE-D-23-27749R2

Dear Dr. Beck,

We’re pleased to inform you that your manuscript has been judged scientifically suitable for publication and will be formally accepted for publication once it meets all outstanding technical requirements.

Kind regards,

Timothy Omara, PhD

Academic Editor

PLOS ONE

Additional Editor Comments (optional):

Reviewers' comments:

Reviewer's Responses to Questions

**Comments to the Author**

1. If the authors have adequately addressed your comments raised in a previous round of review and you feel that this manuscript is now acceptable for publication, you may indicate that here to bypass the “Comments to the Author” section, enter your conflict of interest statement in the “Confidential to Editor” section, and submit your "Accept" recommendation.

Reviewer #2: (No Response)

2. Is the manuscript technically sound, and do the data support the conclusions?

Reviewer #2: Yes

3. Has the statistical analysis been performed appropriately and rigorously? 

Reviewer #2: Yes

4. Have the authors made all data underlying the findings in their manuscript fully available?

Reviewer #2: Yes

5. Is the manuscript presented in an intelligible fashion and written in standard English?

Reviewer #2: Yes

6. Review Comments to the Author

Reviewer #2: I think that the authors have made sufficient revisions and, therefore, I am willing to accept the paper for publication.

7. PLOS authors have the option to publish the peer review history of their article (what does this mean?). If published, this will include your full peer review and any attached files.

Reviewer #2: No

---

## [Editor Report · Acceptance letter]

23 Oct 2023

PONE-D-23-27749R2 

MassWateR: Improving quality control, analysis, and sharing of water quality data 

Dear Dr. Beck:

I'm pleased to inform you that your manuscript has been deemed suitable for publication in PLOS ONE. Congratulations! Your manuscript is now with our production department. 

Kind regards, 

on behalf of

Dr. Timothy Omara 

Academic Editor

PLOS ONE